# The Effect of Consumption of Animal Milk Compared to Infant Formula for Non-Breastfed/Mixed-Fed Infants 6–11 Months of Age: A Systematic Review and Meta-Analysis

**DOI:** 10.3390/nu14030488

**Published:** 2022-01-23

**Authors:** Julie M. Ehrlich, Joseph Catania, Muizz Zaman, Emily Tanner Smith, Abigail Smith, Olivia Tsistinas, Zulfiqar Ahmed Bhutta, Aamer Imdad

**Affiliations:** 1College of Medicine, SUNY Upstate Medical University, Syracuse, NY 13210, USA; ehrlichj@upstate.edu (J.M.E.); cataniaj@upstate.edu (J.C.); zamanm@upstate.edu (M.Z.); 2College of Education, University of Oregon, Eugene, OR 97403, USA; etanners@uoregon.edu; 3Health Science Library, SUNY Upstate Medical University, Syracuse, NY 13210, USA; smithab@upstate.edu (A.S.); tsisitjo@upstate.edu (O.T.); 4Centre for Global Child Health, The Hospital for Sick Children, Toronto, ON M5G 0A4, Canada; zulfiqar.bhutta@aku.edu; 5Division of Pediatric Gastroenterology, Hepatology and Nutrition, Department of Pediatrics, SUNY Upstate Medical University, Syracuse, NY 13210, USA

**Keywords:** animal’s milk, infant formula, anemia, cow’s milk

## Abstract

Many infants do not receive breastmilk for the recommended 2-year duration. Instead, alternative milk beverages are often used, including infant formula and raw animal milk products. The purpose of this systematic review was to summarize the effect of animal milk consumption, compared to infant formula, on health outcomes in non-breastfed or mixed-fed infants aged 6–11 months. We searched multiple databases and followed Cochrane guidelines for conducting the review. The primary outcomes were anemia, gastrointestinal blood loss, weight-for-age, length-for-age, and weight-for-length. Nine studies were included: four randomized controlled trials (RCT) and five cohort studies. All studies, except one, were conducted in high income countries. There was a low certainty of evidence that cow’s milk increased the risk of anemia compared to formula milk (Cohort studies RR: 2.26, 95% CI: 1.15, 4.43, RCTs: RR: 4.03, 95% CI: 1.68, 9.65) and gastrointestinal blood loss (Cohort study RR: 1.52, 95% CI: 0.73, 3.16, RCTs: RR: 3.14, 95% CI: 0.98, 10.04). Additionally, there was low certainty evidence that animal milk consumption may not have a differential effect on weight and length-for-age compared to formula milk. Overall, the evidence was of low certainty and no solid conclusions can be drawn from this data. Further studies are needed from low- and middle-income countries to assess optimal milk type in non-breastfed infants aged 6–11 months.

## 1. Introduction

The World Health Organization (WHO) and United Nations Children’s Fund (UNICEF) recommend exclusive breastfeeding for the first six months of life with the continuation of breastfeeding for up to 2 years or beyond with complementary feeding beginning at 6 months of age [1,2]. However, many infants do not receive breastmilk exclusively through 6 months of age, or breastfeeding might be stopped before the recommended duration of 2 years [3]. According to the Lancet Breastfeeding series, 37% of children aged 6–24 months do not receive breast milk, with variation in rates of 18% in the lower-income countries, 34% in the lower-middle-income countries, and 55% in the upper-middle-income countries [4]. Instead, alternative milk beverages are often used, including infant formula and raw animal milk products [3,4,5]. Cow’s milk contains higher levels of protein (0.9–1.2 g/100 mL in human milk vs. 1.8–2.0 in cow’s milk) and lower lactose levels (7.0 g/100 mL in human milk vs. 4.1 g/100 mL in cow’s milk) [6]. Additionally, human milk has higher iron than cow’s milk (70 mg/ 100 mL vs. 0.07 mg/100 mL) [6]. Due to the differences between human and cow’s milk, the use of cow’s milk in infancy has been associated with gastrointestinal blood loss, iron deficiency anemia, and increased solute load for kidneys [3,7,8,9]. Despite these adverse effects, there are conflicting opinions on the safety of feeding cow’s milk between 6 and 12 months of age. The WHO’s guiding Principles for Feeding Non-breastfed Children 6–24 Months of Age states that feeding animal milk and appropriate complementary foods is a safe choice since iron deficiency provoked by gastrointestinal blood loss resolves by 12 months of age [3,9]. Furthermore, the same effects are not seen if the milk is heat-treated, and iron deficiency can be avoided by using iron supplements or supplementary foods with adequate bioavailability of iron [3,9]. Alternatively, the Dietary Guidelines for Americans (2020) states that infants should not consume cow’s milk before age 12 months as their primary milk drink [10]. 

Infant formula has historically been derived from cow’s milk, but is altered to make it more similar to human breast milk [6]. There have been improvements made to infant formula over the last 20 years that may further increase its benefits compared to cow’s milk, including the addition of oligosaccharides, lactoferrin, and osteopontin, as well as several other micronutrients and functionally active ingredients [6]. Oligosaccharides are nondigestible carbohydrates found in human milk that provide protection against viruses and bacteria [6]. Lactoferrin binds iron and sialic acid which are necessary for growth and development [6]. Osteopontin is a protein necessary for bone development [6]. With these advancements and others in formula through recent years, this review’s objective was to synthesize the most recent research on the effects of the consumption of animal milk compared to infant formula in non-breastfed or mixed breastfed infants aged 6–11 months of age.

Objective: For non-breastfed or mixed-fed (breastmilk and formula) infants 6–11 months of age, is the consumption of animal milk, compared to infant formula, associated with beneficial or adverse outcomes for health and development?

## 2. Materials and Methods

We followed the Cochrane Collaboration’s standard guidelines for this review, and we followed the PRISMA guidelines to report our results. The review’s detailed methods were published in a protocol [11], and the protocol was also pre-registered on the PROSPERO registry (ID: CRD42020210925).

We included individual and cluster randomized trials, quasi-randomized experimental design studies, and prospective and retrospective cohort studies with a control group.

The study population was apparently healthy infants 6–11 months of age who were non-breastfed or mixed fed (breast milk and formula) irrespective of gestational age and birth weight. We excluded studies with participants who have chronic diseases such as bronchopulmonary dysplasia, genetic disorders, aerodigestive problems, or congenital anomalies.

The intervention of interest was the use of animal milk in infants 6–11 months of age. We included studies in which animal milk was the main milk drink as defined by study authors, or more than 50% of the infant’s milk intake was animal milk. We included studies irrespective of whether the animal milk was boiled, pasteurized, or unpasteurized, or if the animal milk was full-fat, reduced-fat, or skim milk. The comparison group in the included studies was formula feeding or mixed feeding (i.e., breastfeeding and formula feeding). We included studies irrespective of the type of formula used; this could include cow’s milk-based formula, partially or extensively hydrolyzed formula, or plant-based formulas such as soy formula. The Food and Drug Administration (FDA) of the United States Federal Food, Drug, and Cosmetic Act (FFDCA) defines infant formula as “a food which purports to be or is represented for special dietary use solely as a food for infants by reason of its simulation of human milk or its suitability as a complete or partial substitute for human milk” [12].

The primary outcomes of interest were: any anemia (dichotomous, as defined by authors); gastrointestinal blood loss (dichotomous, based on stool occult testing); weight-for-age (continuous, kg or Z scores); length-for-age (continuous, cm or Z scores); and weight-for-length Z score. The secondary outcomes were: iron deficiency anemia (dichotomous); serum iron level (continuous); serum ferritin level (continuous); stool hemoglobin concentration (continuous); blood hemoglobin concentration (continuous); serum triglycerides (continuous); diarrhea (dichotomous, defined as >3 loose stools per day); constipation (dichotomous, defined <3 bowel movements per week); pneumonia (dichotomous as defined by authors); allergy (dichotomous IgE-Mediated and non-IgE Mediated and mixed); obesity (dichotomous); overweight (dichotomous); and neurodevelopmental outcomes (continuous). We considered the time of follow-up for these outcomes at 7 months, 9 months, and 12 months as the longest follow up.

We conducted systematic electronic searches on multiple databases, including PubMed, EMBASE, the Cochrane Central Register for Controlled Trials, Web of Science, CINHAL, Scopus, and WHO Global Index Medicus. There were no restrictions applied to the searches based on outcomes, study design, publication status, publication date, or language. The last date of the search was 17 November 2021. Our search strategy for all the databases is shown in Appendix A.

Searches from all the databases were combined in bibliographic software (EndNote) [13], and duplicates were removed. Two authors (either JE, AI, JC, or MZ) screened the titles using the software Covidence [14]. Two authors (JE, JC, or MZ) independently extracted the data from the included studies and compared their findings. Any conflict was resolved by discussion and with the help of the senior author (AI), if needed. The risk of bias was assessed using the Cochrane risk of bias tool-2 (ROB 2.0) [15] for RCTs and using the Cochrane risk of bias in non-randomized studies (ROBINS-I) tool for non-randomized studies [16]. The risk of bias was assessed by two authors for each study included in a pooled analysis.

We reported findings from all included studies in a narrative synthesis and conducted meta-analyses to synthesize evidence across studies quantitatively. Meta-analyses were conducted when data was available from more than one study and clinical and methodological homogeneity was present in the included studies. Dichotomous outcomes were pooled to obtain an average relative risk (RR). For continuous outcomes, we pooled the data to obtain a standardized mean difference (SMD). All study-level and average effect sizes are reported alongside their 95% confidence intervals (CIs). We used the generic inverse variance weighting method for meta-analysis. We used a random-effects model for meta-analysis, given that there might be heterogeneity in effects due to variability in the study populations and interventions used. We analyzed randomized controlled trials and cohort studies separately. We used RevMan [17] software for the statistical analysis.

Statistical heterogeneity in the pooled analysis was assessed using Tau^2^, χ^2^, and I^2^ statistics, and it was considered substantial if the P-value for the Chi^2^ test was less than 0.10, the I^2^ value exceeded 50%, and inspection of forest plots showed substantial variability in the effect of the intervention.

We aimed to assess small study and publication bias using funnel plots and regression tests. However, the number of included studies in the meta-analysis was less than ten, so no testing was performed for publication and small study bias (per the analysis protocol) [11].

We assessed the overall certainty of evidence for the effect of the intervention on each primary outcome and select secondary outcomes using the Grading of Recommendations Assessment, Development, and Evaluation (GRADE) using the software GradePro [18]. We rated the overall body of evidence to certainty level as very low (we have very little confidence in the effect estimate), low (we have limited confidence in the effect estimate), moderate (we have moderate confidence in the effect estimate; the true effect is likely close to the estimate of the effect), or high (we have high confidence that the true effect lies close to that of the estimate of the effect). We present the results of the GRADE assessment in the form of GRADE Evidence Profiles for the primary outcomes and the following secondary outcomes: blood hemoglobin concentration, iron deficiency anemia, constipation, diarrhea, and neurodevelopmental outcomes.

We aimed to compare effects for the following subgroups when possible: age group: age of initiation at 7 months vs. 9 months; country, low- and middle-income country vs. high-income country; type of Feeding: non-breastfeeding vs. mixed feeding; and type of animal milk, cow, goat, buffalo, camel, or sheep. Finally, we considered the following sensitivity analyses: studies with a high overall risk of bias excluded; type of model: and random vs. fixed-effect meta-analysis model.

## 3. Results

### 3.1. Literature Search

The literature search revealed 4340 titles after the exclusion of duplicates. Figure 1 shows the results of the literature search. After screening the full texts of 96 studies for eligibility, we ultimately included nine studies [7,19,20,21,22,23,24,25,26] available in 11 publications (complete list in Appendix A). We excluded 87 studies, and reasons for exclusion for each study can be found in table of excluded studies in Appendix A.

### 3.2. Characteristics of Included Studies

Table 1 and Table 2 display characteristics of include studies for participant and intervention, respectively.

### 3.3. Study Type and Location

Four of the included studies were RCTs [19,20,23,26], and five were prospective observational cohort studies [7,21,22,24,25]. Four studies were conducted in the United States [7,19,25,26], two were from the United Kingdom [21,23], one from Iceland [24], and one from Peru [20]. One study had a multicenter design and took place in 10 countries, including Greece, Spain, Hungary, Ireland, Italy, Portugal, Germany, Chile, Sweden, and Austria [22].

### 3.4. Study Population

The studies’ median sample size was 133 participants with a range from 15 participants [20] to 971 participants [21]. The mean age of initiation of cow’s milk feeding was 6.95 months of age with the range of 3.6 months [7] to 10 months of age [22].

### 3.5. Type of Animal Milk and Comparators

All studies compared cow’s milk feeding as the animal’s milk group and formula feeding only as the comparison. Five studies [7,19,20,23,26] used pasteurized homogenized cow’s milk. One study [26] used fortified cow’s milk. Another study used heat-treated cow’s milk [7]. For the comparison groups, six studies [7,19,20,23,25,26] used cow’s milk-based infant formulas. One study [7] used Enfamil as the formula group. Two studies [19,20] used Similac with iron and Carnation follow-up formula as the two formula groups. One study [26] used a specially prepared formula similar in composition to commercially prepared Enfamil, except it contained no added iron. Two studies used iron-supplemented standard cow milk formulas [23,25]. Three of the studies were non-specific about the types of milk or formula used in studies [22,23,24].

### 3.6. Studies with Multiple Intervention Arms and Missing Data

Five studies had multiple intervention arms that we combined to obtain a single pairwise comparison [7,20,21,23,28]. For one study [7] we combined whole milk and heat-treated cow’s milk for males and females. For three studies [19,20,23], we combined the formula groups. For two studies [20,21], we used standard deviations from studies with similar populations and sample sizes. Annex 4 gives further details on studies where data was converted for this review.

### 3.7. Co-Interventions

Four studies included co-interventions. One study had infants in the cow’s milk feeding group eat iron-fortified cereal daily [19], while another study had both the cow’s milk feeding group and the formula feeding group consume iron-fortified cereal daily [25]. One study had all infants take a supplement that contained ascorbic acid, iron from ferrous sulfate, and fluoride from sodium fluoride [7]. Another study had infants consuming cow’s milk take a supplement of ascorbic acid and fluoride in the form of sodium fluoride. At the same time, the formula group received a daily supplement of only fluoride [26].

### 3.8. Confounding Variables Included in Analysis

Four studies (one RCT and three observational studies) included confounding variables in their analysis [21,22,23,24]. One study [23] from the United Kingdom randomized infants from the Indian subcontinent separately due to the high risk for iron deficiency in this population. Additionally, it adjusted for the following variables: breastfeeding, the median duration of breastfeeding, first child, mothers’ education, and non-manual social class for neurodevelopmental outcomes expressed as Bayley psychomotor development index (PDI). A prospective cohort study [21] adjusted results for: maternal education, smoking in pregnancy, and parity as confounding variables for mean weight. Another cohort study [22] used multiple regression analysis to test associations between hemoglobin, serum ferritin, mean corpuscular volume, transferrin saturation, transferrin as dependent variables, demographic and dietary factors, growth, and morbidity as independent variables. One cohort study [24] used a linear model which was adjusted for gender, birth weight, and length of exclusive breastfeeding. Two observational studies did not adjust for any confounding variables [7,25].

### 3.9. Effects of Interventions

In the section below, we report the meta-analysis and GRADE analysis results for each primary outcome and select secondary outcomes at the longest follow-up. The results for the primary outcomes at other durations of follow up such as at 7, 9, and 12 months are available in Table 3. Table 4 shows GRADE evidence profiles for the primary outcomes and select secondary outcomes. All of the included studies contributed data to an outcome for meta-analysis except two cohort studies [21,22] where the data were reported in a way that could not be meta-analyzed. The results for these studies were reported individually in this review.

### 3.10. Primary Outcomes

#### 3.10.1. Anemia at the Longest Follow-Up

Two RCTs [23,28] and two cohort studies [24,25] reported data on anemia. Each study used slightly different definitions of anemia (please see Appendix A for definitions). Data from the two cohort studies included 327 participants, with 155 in the milk feeding group and 172 in the formula feeding group. Data from the two RCT studies included 209 participants, with 60 in the milk feeding group and 149 participants in the formula feeding group. Three of the studies had the longest follow-up time as 12 months [24,25,28] while one study had the longest follow-up time as 18 months [23]. The results showed, with a low certainty of evidence, that cow’s milk as the main milk drink leads to an increase in anemia when compared to formula feeding in infants 6–11 months of age (Cohort studies RR: 2.26, 95% CI: 1.15, 4.43, No. of studies: 2; *p* = 0.02, I^2^ = 0%, Grade certainty: Low; Randomized controlled trials: RR: 4.03, 95% CI: 1.68, 9.65, No. of studies: 2; *p* = 0.002, I^2^ = 0%; Grade certainty: Low: Figure 2). We downgraded the GRADE certainty for risk of bias (one observational study had a ‘high’ risk of bias [25] and another [24] had ‘moderate’ risk of bias and one RCT [19] had “some concerns”) and indirectness (all the studies were conducted in high-income countries) (Table 4). 

##### Subgroup and Sensitivity Analysis

The a priori subgroup analyses based on the type of country (Low and middle-income country vs. high-income country), type of animal milk (cow, goat, buffalo, camel, or sheep), and type of feeding in the comparison group (non-breastfeeding vs. mixed feeding) were not performed as all the studies were conducted in high-income countries and used cow’s milk as the animal milk. A subgroup analysis based on the age of initiation did not show any significant difference between the groups at 7 months and 9 months of age (*p*-value for subgroup difference 0.61, Appendix A). Sensitivity analysis based on the type of model used showed the same results for the fixed vs. random model of meta-analysis. One of the cohort studies [25] had a high risk of bias due to lack of adjustment of confounding variables. Exclusion of this study from meta-analysis of the cohort studies changed the summary estimate and the statistical significance of the summary estimate (RR 6.28, 95% CI 0.33, 119.77). One RCT had “some concerns” for risk of bias [19] and exclusion of this study from meta-analysis of RCTs did not change the direction or the statistical significance of the summary estimate (RR 3.77, 95% CI 1.52, 9.36).

#### 3.10.2. Gastrointestinal Blood Loss

Two studies [7,26] reported data on gastrointestinal blood loss. One study was a cohort study [7], and the other was an RCT [26]. The cohort study included a total of 81 participants, with 60 participants in the milk feeding group and 21 participants in the formula feeding group. The RCT included 43 participants, with 21 in the milk feeding group and 22 in the formula feeding group. Both the studies quantified gastrointestinal blood loss using the guaiac stool test. The cohort study had the longest follow-up time of 6.54 months [7], while the RCT had the longest follow-up time of 8.28 months [26]. The results showed a very low to low certainty of evidence that cow’s milk leads to increased gastrointestinal blood loss (Cohort study RR: 1.52, 95% CI: 0.73, 3.16, No. of studies: 1; *p* = 0.27; Grade certainty: very Low: Randomized controlled trial: RR: 3.14, 95% CI: 0.98, 10.04, No. of studies: 1; *p* = 0.05; Grade certainty: Low: Figure 3). We downgraded the GRADE certainty to very low for cohort study because of the risk of bias (cohort study had ‘high’ risk of bias), imprecision (wide confidence intervals and they included a null effect), and indirectness (study was conducted in high-income country) (Table 4). We downgraded the certainty to low for the RCT because of ‘some concerns’ for risk of bias and indirectness (the study was conducted in high-income country) (Table 4). 

#### 3.10.3. Weight for Age

Three RCTs reported weight for age and included 556 participants, with 194 in the animal milk feeding group and 362 in the formula feeding group [20,23,28]. Two studies had a follow-up time of 12 months [20,27]. One study had a follow-up time of 18 months [23]. All three studies included in the meta-analysis were RCTs [20,23,28]. Two [23,28] of the RCTs had ‘some concerns’ for risks of bias, and the other a ‘high’ risk of bias [20]. The pooled results showed no evidence that cow’s milk compared to formula feeding had an effect on weight for age (Randomized Controlled Trials: SMD: −0.02, 95% CI: −0.26, 0.21, No. of studies: 3; *p* = 0.84, I^2^ = 19%, Figure 4, GRADE certainty: low). We downgraded the GRADE certainty to low due to the risk of bias and indirectness (all but one of the included studies were conducted in high-income countries) (Table 4). 

##### Subgroup and Sensitivity Analysis

The subgroup analysis based on the age of initiation of animal milk did not show any significant difference between the age groups of 7 and 9 months (*p*-value for subgroup difference: 0.42, Appendix A).

A sensitivity analysis based on a fixed-effect model did not change the direction or statistical significance of the summary estimate (SMD 0.00, 95 % CI −0.18, 0.18). Removal of one study [20] with a high risk of bias did not change the direction or statistical significance of the summary estimate (SMD 0.11; 95% CI −0.03, 0.25).

#### 3.10.4. Length for Age

Two RCTs reported length for age and included a total of 529 participants, with 185 in the cow milk feeding group and 344 in the formula feeding group [23,27]. One study had a follow-up time of 12 months [27]. One study had a follow-up time of 18 months [23]. The results showed no evidence that cow’s milk compared to formula feeding had an effect on length (Randomized Controlled Trials: SMD: 0.07, 95% CI: −0.15, 0.30, No. of studies: 2; *p* = 0.51, I^2^ = 17%, Figure 5, Grade certainty: low). We downgraded the GRADE certainty of evidence to low because of the risk of bias (one of the two studies had ‘some concerns’ for risk of bias) and indirectness (all studies were conducted in high-income countries) (Table 4). 

#### 3.10.5. Weight for Length

No studies reported data on weight for length.

### 3.11. Secondary Outcomes

We describe the results of selected secondary outcomes for which the GRADE analysis was conducted (Table 4). The results for all the secondary outcomes at the longest follow-up are available in Table 5.

#### 3.11.1. Blood Hemoglobin Concentration

Three RCTs [23,26,28] and two cohort studies [7,24] reported data on hemoglobin concentration in the blood. The two cohort studies had a total number of 246 participants, with 148 in the milk feeding group and 98 in the formula feeding group. Three RCTs had a total number of 250 participants, with 82 in the milk feeding group and 168 in the formula feeding group. The results showed low certainty evidence that use of animal milk reduces the hemoglobin concentration in blood compared to formula (Cohort studies SMD = −0.37, 95% CI: −0.78, 0.05, No. of studies: 2; *p* = 0.09, I^2^ = 52%; Grade certainty: Low: Randomized Controlled Trials: SMD −0.32, 95% CI: −0.59, −0.05, No. of studies: 3; *p* = 0.02, I^2^ = 0%, Grade certainty: Low, Figure 6). We downgraded the GRADE certainty to low because of the risk of bias and indirectness (Table 4). NA refers to no I^2^ because there is only 1 study available for that outcome. 

#### 3.11.2. Iron Deficiency Anemia

Two cohort studies reported iron deficiency anemia with 327 participants, 155 in the cow’s milk group and 172 in the formula group [24,25]. The pooled results showed a low certainty evidence that the use of animal milk increases the risk of iron deficiency anemia (Cohort studies: RR: 2.26, 95% CI: 1.15, 4.43, No. of studies: 2; *p* = 0.02, I^2^ = 0%, Figure 7). We downgraded the evidence for the risk of bias and indirectness (Table 4).

#### 3.11.3. Neurodevelopmental Outcomes

One RCT [23] reported data on neurodevelopmental outcomes and the results did not show a significant difference in neurodevelopmental outcome for PDI (psychomotor developmental index) scores (SMD 0.18, 95% CI −0.02, 0.37, *p* = 0.10, No. of study: 1, total participants 428, GRADE certainty: low) or MDI (Mental developmental index) scores (SMD 0.16, 95 % CI −0.03, 0.36, *p* = 0.10, No. of study: 1, total participants 428, GRADE certainty: low).

#### 3.11.4. Gut Health (Diarrhea and Constipation)

Data from one cohort study [25] showed very low certainty evidence that the use of animal milk might increase the risk of diarrhea (RR 1.86, 95% CI: 1.05, 33.10) and constipation (RR 3.31, 95% CI: 0.89, 12.37) (Table 4).

#### 3.11.5. Other Outcomes

Four studies [23,24,26,28] reported data on serum ferritin level, one cohort study and three RTCs. The cohort study had a total number of 165 participants, with 87 in the milk feeding group and 78 in the formula feeding group. The RCTs had a total number of 406 participants, with 141 in the milk feeding group and 265 in the formula feeding group. The results showed a decrease of serum ferritin level in the animal milk group compared to formula-fed infants (Cohort study SMD: −0.81, 95% CI: −1.13, −0.49, No. of studies: 1 *p* < 0.00001; Randomized controlled trial: SMD: −0.30, 95% CI: −0.94, 0.34, No. of studies: 3 *p* = 0.35, Figure 8). 

Two studies [26,28] reported data on hemoglobin concentration in the stool with a total of 223 participants, with 93 in the cow’s milk group and 135 in the formula feeding group. The pooled results did not show a significant difference in the stool hemoglobin concentration between the two groups (SMD: 0.22, 95% CI: −0.16, 0.59, *p* = 0.26, No. of studies: 2; I^2^ = 41%, Figure 9). 

One study [26] reported data on blood iron level with serum iron of 4.60 mg/dL in the cow’s milk feeding group and 4.49 mg/dL in the formula feeding group (not significant). One study [25] reported data on triglyceride level. The cow’s milk feeding group had a triglyceride level of 107.5 mg/dL, and the formula feeding group had a triglyceride level of 117.5 mg/dL (not significant). One study [25] reported data on allergies but reported it as the level of IgE (continuous variable) rather than as a number of infants presenting with allergies (dichotomous value). IgE level in the cow’s milk group was 8.3 IU (international unit) and the IgE level in the formula group was 11.1 IU (not significant). No study reported data on pneumonia, obesity, or overweight.

Male et al. 2001 [22] was not included in the meta-analysis because their data was not presented in a way that would allow us to compare cow’s milk to formula directly. Instead, the authors presented the effects of cows’ milk feeding as a function of time. They reported that the duration of feeding cows’ milk decreased hemoglobin levels by 2 g/L for every month fed cows’ milk. Additionally, authors found that each month of cow’s milk feeding increased anemia by 23%, iron deficiency by 18%, and iron deficiency anemia by 39% (*p* < 0.001; *p* < 0.01; *p* < 0.001, respectively).

Hopkin’s et al. 2015 [21] was also not included in the meta-analysis due to differences in reporting of data. The study aimed to assess the weight gain in infants who were fed cow’s milk, formula milk, or breastmilk. The participants were followed for a maximum of 10 years. There was no difference in weight gain velocity during infancy; however, infants who were fed high volumes of cow’s milk (>600 mL/day) in early infancy had higher weight and height gain after infancy. The study had more than 30% loss to follow up, and the study did not adjust for missing data, so we had low confidence in the reported results.

## 4. Discussion

### 4.1. Summary of Main Results

This systematic review and meta-analysis evaluated the effect of animal milk vs. infant formula as the main milk drink for non-breastfed/mixed fed infants 6–11 months of age. Results from this evidence synthesis suggest that there is low certainty evidence that the use of animal milk compared to infant formula may increase the risk of anemia and blood loss in the gastrointestinal tract and decrease the levels of blood hemoglobin and serum ferritin. Furthermore, low certainty evidence showed that use of animal milk may not have a differential effect on weight and length for age compared to formula milk. There was limited data available for neurodevelopmental outcomes and adverse events, such as constipation and diarrhea, so no conclusive statements could be made in this regard.

### 4.2. Overall Completeness and Applicability of Evidence

This review summarized evidence from both RCTs and observational cohort studies. We included nine studies comprising 2536 participants; however, data were not available for all outcomes in all the included studies. There were not enough studies to perform all the a priori subgroup analyses. Therefore, no conclusions can be drawn for the differential effects of animal milk based on country, or type of animal milk. Additionally, because all the included studies used cow’s milk for the animal milk, findings from this review cannot be generalized to other types of animal’s milk such as goat or buffalo milk. Further, given limited variability in the literature, we could not examine any differential effects based on the treatment of cow’s milk, such as heated vs. non-heated, pasteurized vs. non-pasteurized, or diluted vs. non-diluted milk. The subgroup analyses based on age did not have enough studies to make any conclusive statements about the differential effect of animal milk given at different ages (Table 3).

Overall, findings from this review suggest that the use of cow’s milk between 6–11 months of age may increase the risk of anemia during infancy. All the measured outcomes related to anemia in this review showed a negative association with cow’s milk use compared to infant formula. And the effect seems homogenous as statistical heterogeneity measured based on I^2^ was non-significant in almost all the anemia-related outcomes. This finding is interesting to note as at least three studies either fortified the cow’s milk or supplied additional iron with the help of fortified complementary foods [7,19,26]. However, these three studies did not contribute data to all the anemia-related outcomes, and their relative contribution to overall summary estimates varied from outcome to outcome. Therefore, we cannot comment with great confidence if the risk of anemia could be averted with additional supplementation by fortifying cow’s milk and complementary feedings and future studies might be needed to further elaborate on this aspect of the intervention. The proposed mechanisms by which cow’s milk may increase anemia’s risk are related to decreased amount of iron in cow’s milk, decreased bioavailability of iron, and potentially increased blood loss from the gastrointestinal tract [3].

The use of cow’s milk does not seem to have a differential effect on growth outcomes, although the certainty of evidence for growth outcomes was low. One RCT [23] reported data for the neurodevelopmental outcomes, and no difference was found between the two groups for PDI or MDI scores so that no solid conclusion can be drawn about the beneficial or adverse effect of cow’s milk in this regard. One cohort study [25] reported data on an increased risk of constipation and diarrhea in the cow’s milk group compared to formula milk. However, the number of participants with these outcomes were small, and no solid conclusion can be drawn that cow’s milk increases the risk of constipation or diarrhea compared to formula milk.

### 4.3. Certainty of Evidence

Overall, the number of included studies in each analysis was small. As a result, most of the outcomes received a certainty of evidence rating as low or very low. We downgraded the certainty of evidence for most of the outcomes for indirectness because all but one [20] of the included studies were conducted in high-income countries and may not represent low- and middle-income populations. For instance, the risk of anemia might be higher in low- and middle-income countries where the use of animal milk is higher in infants, and there is an increased incidence of diarrhea illness leading to less oral intake and increase losses from the gastrointestinal tract [29].

Two observational cohort studies did not adjust for confounding variables and were rated at high risk of bias [7,25]. One of the RCTs was at high risk of bias due to inadequate methods of randomization [20]. We downgraded the certainty ratings for the risk of bias where applicable.

### 4.4. Potential Bias in the Review Process

We followed the standard guidelines of the Cochrane Collaboration to conduct this review. We adopted a broad search strategy, used multiple databases, and examined 4340 titles and abstracts, including published and ongoing studies. We performed our analysis according to an a priori plan, and the protocol was published in a peer-reviewed journal [11]. We analyzed the data separately for randomized controlled trials and cohort studies. Most of the analyses from observational cohort studies mirrored the evidence from randomized controlled trials, suggesting a true increased risk of anemia with use of cow’s milk in non-breastfed infants greater than six months of age. Some of the included did not provide the data needed for meta-analysis. We adopted the standardized methods to use data from other published studies in case of missing data such as standard deviations for continuous outcomes. Some studies had more than two study groups and we combined certain groups to avoid double counting of the control group. We were transparent about any decisions made related to missing data and data analysis (Annex 4).

### 4.5. Agreement and Disagreement with Other Studies or Review

To the best of our knowledge, no prior study has attempted a meta-analysis for the use of animal milk vs. formula milk in infants 6–11 months. One review published in 2015 included children up to 3 years of age and similarly reported an increased risk of iron deficiency anemia with the use of cow’s milk compared to follow-on formula [30]. The pooled results from this review showed a three times higher risk of iron deficiency anemia for infants consuming cow’s milk compared with those drinking follow-on formula with a relative risk of 3.3 [30]. In these four studies 25–38% of infants consuming cow’s milk developed IDA compared with 2–15% of those fed with iron fortified formula. This study was less restrictive in age of initiation of milk feeding and followed infants until 3 years of age [30]. Similarly, they found the quality of evidence to be low [30]. Additionally, they were unable to draw conclusions about infant growth between cow’s milk and infant formula as in our study [30]. Recent guidelines for healthy American infants recommend avoiding the use of cow’s milk in infants less than 12 months of age, based on a qualitative synthesis of the evidence [10]. Our review added to the literature the qualitative and quantitative synthesis of the data for use of animal milk during 6–11 months of age.

### 4.6. Implication for Practice

Use of cow’s milk compared to formula milk in infants 6–11 months of age in non-breastfed/mixed-fed infants seems to increase the risk of anemia. However, it is important to note that all the included studies in this review were conducted in high-income countries except for one conducted in Peru [20], an upper-middle-income country. Thus, the generalization of these results to low- and middle-income settings should be considered with caution. Furthermore, a standardized infant formula may not be readily available in low- and middle-income countries and might be expensive compared to cow’s milk [31,32]. Other strategies to mitigate the anemia in this age group, such as fortified complementary foods should be studied further in non-breastfed infants [3].

### 4.7. Implication for Research

Most of the studies included in this review were relatively old, and there was a paucity of data from the last two decades. Moreover, only one study was from an upper-middle-income country, and there was a paucity of data from low and middle-income countries. Future research might be warranted to examine the effect of animal milk in low- and middle-income countries and to assess if there is a difference in the type of animal milk and the age when animal milk is first introduced. Furthermore, future studies are needed to assess if the risk of anemia with cow’s milk could be reduced with the fortification of cow’s milk and complementary feeding with iron. Furthermore, future studies should assess the risk of anemia when the cow’s milk is pasteurized, heated, or diluted.

## 5. Conclusions

Low certainty evidence showed that feeding cow’s milk to infants 6–11 months of age as the main milk drink, as opposed to formula, seems to increase the risk of anemia and indices of anemia, including iron deficiency anemia and decreased blood hemoglobin and ferritin. There was no differential effect of cow’s milk on weight or length compared to infant formula based on low certainty of evidence. Limited data were available for the outcome of neurodevelopment and adverse effects such as diarrhea and constipation, and no solid conclusions could be drawn for these outcomes. Most of the available studies were conducted in high-income countries, and future studies are needed from low- and middle-income countries to assess the optimal milk-type use in non-breastfed/mixed fed infants 6–11 months of age.

## Figures and Tables

**Figure 1 nutrients-14-00488-f001:**
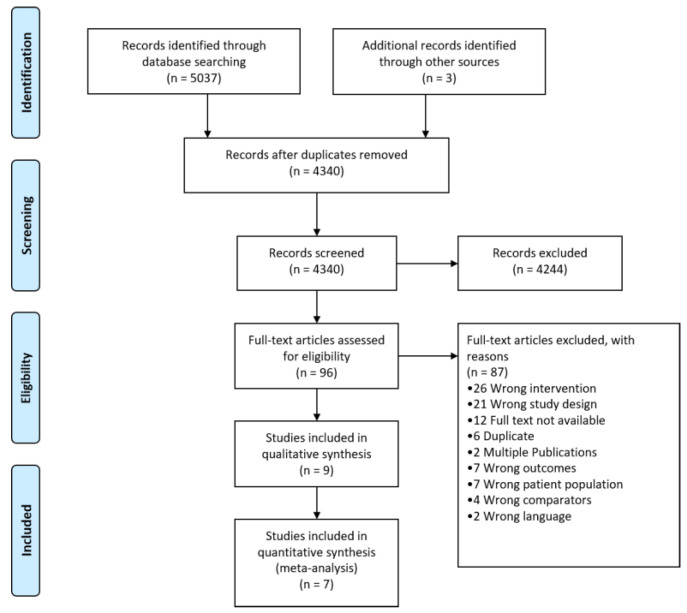
PRISMA Flow Diagram showing results of the literature search.

**Figure 2 nutrients-14-00488-f002:**
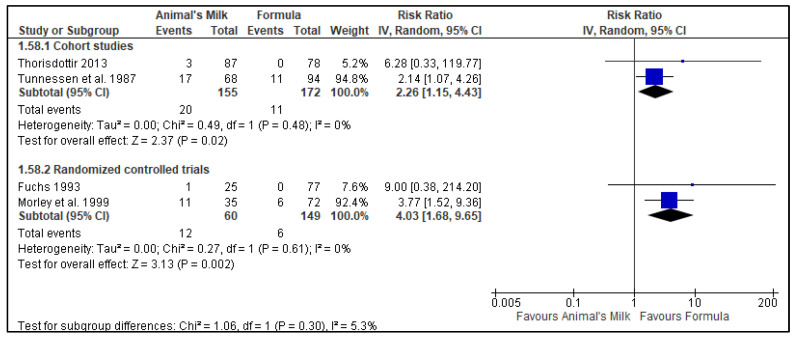
Effect of animal’s milk vs. formula milk intake in infants 6–11 months of age on Anemia. Footnotes: The figure shows results of meta-analysis for use animal milk vs. formula milk for non-breastfed infants based on study type. Only subtotals were calculated as we had decided a priori to pool data from observational studies and randomized controlled trials separately. The direction of effect from both the studies mirror each other and data from randomized trials seems to be confirmatory of effect seen from cohort studies. Abbreviations: CI, Confidence interval; and IV, Inverse variance.

**Figure 3 nutrients-14-00488-f003:**
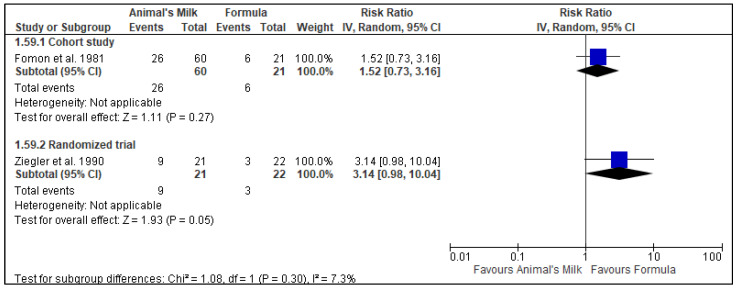
Effect of animal’s milk vs. formula milk intake in infants 6–11 months of age on gastrointestinal blood loss. Footnotes: The figure shows results of meta-analysis for use animal milk vs. formula milk for non-breastfed infants based on study type. Only subtotals were calculated as we had decided a priori to pool data from observational studies and randomized controlled trials separately. Number of included studies were small and confidence interval around the summary estimate were wide. Abbreviations: CI, Confidence interval; and IV, Inverse variance.

**Figure 4 nutrients-14-00488-f004:**
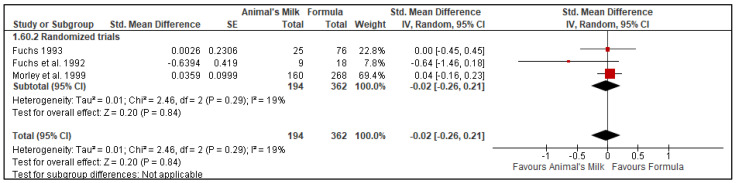
Effect of animal’s milk vs. formula milk intake in infants 6–11 months of age on weight for age. Footnotes: The figure shows results of meta-analysis for use animal milk vs. formula milk for non-breastfed infants based on study type. All the studies were randomized controlled trials. Abbreviations: CI, Confidence interval; IV, Inverse variance, SE: standard error.

**Figure 5 nutrients-14-00488-f005:**
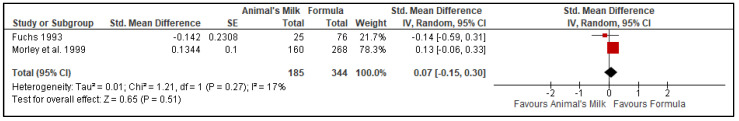
Effect of animal’s milk vs. formula milk intake in infants 6–11 months of age on length for age. Footnotes: The figure shows results of meta-analysis for use animal milk vs. formula milk for non-breastfed infants based on study type. All the studies were randomized controlled trials. Abbreviations: CI, Confidence interval; IV, Inverse variance; and SE, standard error.

**Figure 6 nutrients-14-00488-f006:**
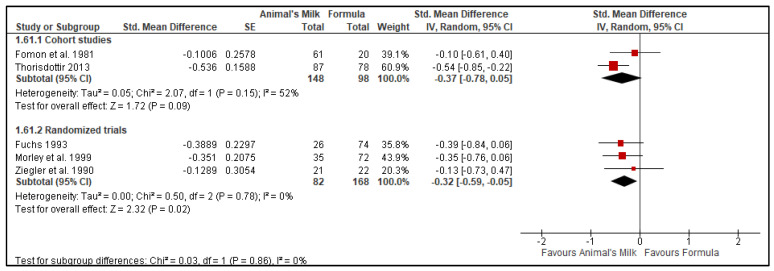
Effect of animal’s milk vs. formula milk intake in infants 6–11 months of age on blood hemoglobin level. Footnotes: The figure shows results of meta-analysis for use animal milk vs. formula milk for non-breastfed infants based on study type. Only subtotals were calculated as we had decided a priori to pool data from observational studies and randomized controlled trials separately. Abbreviations: CI, Confidence interval; IV, Inverse variance; and SE, standard error.

**Figure 7 nutrients-14-00488-f007:**
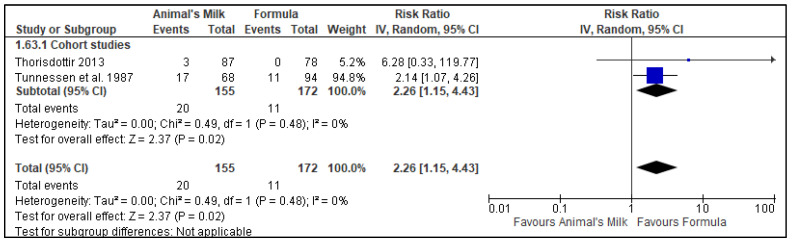
Effect of animal’s milk vs. formula milk intake in infants 6–11 months of age on iron deficiency anemia. Footnotes: The figure shows results of meta-analysis for use animal milk vs. formula milk for non-breastfed infants based on study type. Both the included studies were cohort studies. Abbreviations: CI, Confidence interval; IV, Inverse variance; and SE, standard error.

**Figure 8 nutrients-14-00488-f008:**
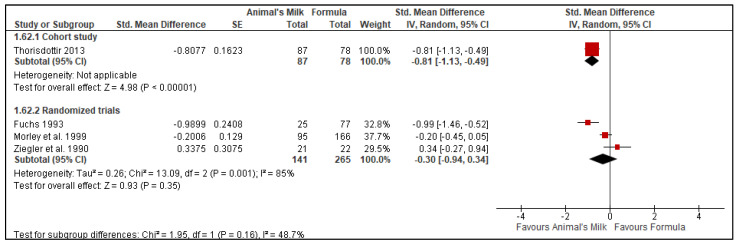
Effect of animal’s milk vs. formula milk intake in infants 6–11 months of age on serum ferritin level. Footnotes: The figure shows results of meta-analysis for use animal milk vs. formula milk for non-breastfed infants based on study type. Only subtotals were calculated as we had decided a priori to pool data from observational studies and randomized controlled trials separately. Abbreviations: CI, Confidence interval; IV, Inverse variance; and SE, standard error.

**Figure 9 nutrients-14-00488-f009:**
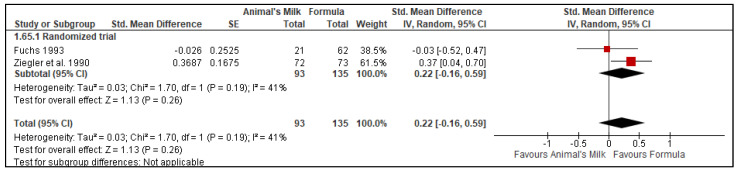
Effect of animal’s milk vs. formula milk intake in infants 6–11 months of age on hemoglobin concentration in the stool. Footnotes: The figure shows results of meta-analysis for use animal milk vs. formula milk for non-breastfed infants based on study type. Both the included studies were randomized controlled trials. Abbreviations: CI, Confidence interval; IV, Inverse variance; and SE, standard error.

**Table 1 nutrients-14-00488-t001:** Participant characteristics in the included studies.

Author	Type of Study	Country	Number of Participants in Study	Inclusion Criteria	Age Initiation of Animal Milk Feedings (Months)
Fomon 1981 [7]	Observational	United States	81	Infants with birth weights >450 gm within four days of 112 days of age	3.6 months
Fuchs 1993, 1993, 1996 [19,27,28]	RCT	United States	104	Healthy, full term, exclusively bottle-fed infants	6 months
Ziegler 1990 [26]	RCT	United States	52	Full term infants with birth weights of 2500 g	5.5 months
Tunnessen 1987 [25]	Observational	United States	169	Infants previously been fed an iron-supplemented proprietary cow milk formula from birth; no whole cow milk before 6 months of age; born at >38 weeks’ gestation, no underlying systemic disease or prior hospital admissions, with a mother who was at least 16 years of age	6 months
Morley 1999 [23]	RCT	United Kingdom	493	Healthy infants born at >36 weeks’ gestation, weighing > 2500 g, and either singletons or sole survivors from a multiple pregnancy	9 months
Thorisdottir 2013 [24]	Observational	Iceland	165	Icelandic parents, singleton birth; gestational length of 37–41 weeks, birth weight within the10th and 90th percentiles, no birth defects or congenital long-term diseases; early and regular antenatal care of the mother.	3.3% received whole milk at 6 months, 40% at 9 months and 56% at 12 months
Male 2001 [22]	Observational	Greece, Spain, Hungary, Ireland, Italy, Portugal, Germany, Chile, Sweden, Austria	488	Birthweight 2500 g, gestational age 37 weeks, single birth, Caucasian origin, no language barrier with parents, known father, and high probability of successful participation for 36 mo.	10 months
Fuchs 1992 [20]	RCT	Peru	15	Infants 6–12 months old, recovering from malnutrition; free of diarrhea, parasites and other apparent infections; were gaining weight at an appropriate rate for their height age; free of edema, skins lesions or other signs of specific nutrient deficiencies and had serum albumin levels of at least 3.4 g/dL.	7.5 months
Hopkins 2015 [21]	Observational	United Kingdom	925	Resident in a geographically defined area of South-West England; expected date of delivery between April 1991 and December 1992; singleton children born at term with dietary information at 8 months of age.	Data is available based on what the child was consuming at 8 months of age

**Table 2 nutrients-14-00488-t002:** Treatment characteristics in the included studies.

Author	Type of Milk	Amount of Milk	Frequency of Milk Drinking	Type of Comparator (Formula, Mixed Feeding)	Amount of Formula	Frequency of Formula Drinking	Fortification/Measured Iron Levels	Co-Interventions	Funding Sources
Fomon 1981 [7]	Cow’s milk ^1^	Ad libitum	Daily	Cow’s milk based infant formula ^2^	Not stated	Ad libitum	Enfamil provided 1.5 mg iron/L, whereas cow milk provided only trace amounts of iron	All infants received daily 1.0 mL of a solution that provided 50 mg ascorbic acid, 12 mg iron from ferrous sulfate, and 0.5 mg fluoride from sodium fluoride.	United States public Health Service grant 1 P01 HD 07578 and a grant-in-aid from National Dairy Council
Fuchs 1993, 1993, 1996 [19,27,28]	Cow’s milk ^3^	7 months = 810 mL, 12 months = 720 mL	Daily	Cow’s milk based infant formula ^4^	7 months = 810 mL, 12 months = 720 mL	Daily	The mothers in the WCM + C (cereal) group were additionally provided with iron-fortified rice, oat or mixed rice–oat cereal and counseled to feed their infants 135 mL (9 tbsp) cereal/d mixed in formula, milk, or water (but not in juice) to achieve the recommended dietary allowance (RDA) of iron of 10 mg (11). The mothers of infants in the formula groups were not given specific instructions about the use of infant cereal or other supplemental foods. Iron composition of 4 groups: WCM + Cereal (before supplementation): 0.38 mg/MJ FUF2: 4.54 mg/MJ FUF1: 4.30 mg/MJ IF: 4.30 mg/MJ	The infants in the cow’s milk group were supplied with dry iron-fortified infant cereal throughout the study period.	Carnation Nutritional Products
Ziegler 1990 [26]	Cow’s milk ^5^	Not stated	Not stated	Cow’s milk based infant formula ^6^	Not stated	Not stated	The measured iron concentration of Enfamil formula without added iron: 0.83 mg/L. Iron concentration of cow milk, determined on several occasions by atomic absorption spectrophotometry after dry ashing, averaged 0.98 mg/L.	Infants fed cow milk received a daily supplement of 35 mg ascorbic acid and 0.25 mg fluoride in the form of sodium fluoride. The formula group received a daily supplement of 0.25 mg fluoride	U.S. Public Health Service grant No. HD 07578 and by grants from the National Dairy Council and Ross Laboratories.
Tunnessen 1987 [25]	Whole cow’s milk (non-specific) ^7^	Not Stated	Daily	Cow’s milk based formula ^8^	Not Stated	Daily		Parents were encouraged to feed iron-fortified cereal throughout the study period.	Wyeth Laboratories
Morley 1999 [23]	Cow’s milk ^9^	Not stated	Daily	Cow’s milk based formula ^10^	Not stated	Daily	Milk: estimated to contain 0.05 mg iron/litre Formula containing 0.9 mg iron/litre Identical formula with 1.2 mg iron/litre as ferrous sulphate	None	Wyeth Laboratories
Thorisdottir 2013 [24]	Cow’s milk (non-specific)	332.5 mL/day	Per Day	Follow-on formula (mainly, non-specific)	378.3 mL/day	Per Day	Cow Milk: Median 3.5 mg/day Formula: 7.9 mg/day	No information	Icelandic Research Council (050424031) and The Icelandic Research Fund for Graduate Students (080740008), University of Iceland Research Fund and Landspitali—University Hospital Research fund
Male 2001 [22]	Cow’s milk (non-specific)	Not stated	Not stated	Formula (non-specific)	Not stated	Not stated	Not given	No information	Euronut, a concerted action of the European Union, and by a grant from the Austrian Ministry of Science
Fuchs 1992 [20]	Whole cow’s milk ^3^	219 mL/kg/day	ad libitum	Cow’s milk based infant formula ^11^	219 mL/kg/day	ad libitum	Not given	None	Carnation Nutritional Products
Hopkins 2015 [21]	Cow’s milk (non-specific)	<600 mL cow milk/day (CMlow); >600 mL cow milk/day (CMhigh)	daily	Formula (non-specific) ^12^	<600 mL formula milk/day (FMlow); >600 mL formula milk/day (FMhigh)	Daily	Not given	No information	Wyeth Nutrition

^1^ “Whole cow milk” designates pasteurized, homogenized cow milk obtained from a local dairy; “Heat-treated cow milk” designates whole cow milk prepared by the manufacturer of Enfamil at the same time as the batch of Enfamil used in the study and using identical time and temperature treatment. ^2^ Enfamil. ^3^ WCM signifies pasteurized homogenized cow milk obtained from a local dairy. ^4^ A ready-to feed infant formula signifies Similac with Iron or one of two ready-to-feed follow-up formulas, an investigational formula, or Carnation Follow-up Formula. ^5^ Locally purchased pasteurized, homogenized whole cow milk fortified with vitamin D, 400 lUlL. ^6^ Specially prepared formula similar in composition to commercially prepared Enfamil, except that it contained no added iron. The measured iron concentration of this formula was 0.83 mg/L. The protein was unmodified (Le. casein predominant) cow milk protein. ^7^ Parents were given coupons to buy whole cow’s milk. ^8^ Iron-supplemented proprietary cow milk formula. ^9^ Pasteurized. ^10^ With 0.9 mg of iron/ liter OR identical formula with 1.2 mg iron/ liter as ferrous sulfate. ^11^ Carnation Follow-up Formula, Carnation Nutritional Products, Glendale, California (FUF), Similac with iron, Ross Laboratories, Columbus, Ohio (IF). ^12^ Formula group defined as formula with or without some BM and/or cow milk.

**Table 3 nutrients-14-00488-t003:** Primary outcome results at different timepoint of follow up.

Outcome	Time Point	No. of Studies	Study Type	Relative Risk	95% CI
Anemia	9 months	1	RCT	0.59	0.03, 11.92
	12 months	1	Cohort	2.26	1.15, 4.43
		2	RCT	9.00	0.38, 214.20
Gastrointestinal Blood Loss	7 months	1	Cohort	1.52	0.73, 3.16
		1	RCT	2.78	0.83, 9.25
	9 months	1	RCT	3.14	0.98, 10.04
Weight-for-age	12 months	1	RCT	0.00	−0.45, 0.45
Length-for-age	12 months	1	RCT	−0.14	−0.59, 0.31

**Table 4 nutrients-14-00488-t004:** GRADE Evidence Profile for Certainty Assessment of primary outcomes and selected secondary outcomes.

Certainty Assessment							№ of Patients		Effect		Certainty
№ of Studies	Study Design	Risk of Bias	Inconsistency	Indirectness	Imprecision	Other Considerations	Animal Milk	Infant Formula	Relative (95% CI)	Absolute (95% CI)	
Anemia at longest follow up-Randomized Controlled Trials											
2	randomised trials	serious ^a^	not serious ^b^	serious ^c^	not serious ^d^	none	12/60 (20.0%)	6/149 (4.0%)	RR 4.03 (1.68 to 9.65)	122 more per 1000 (from 27 more to 348 more)	⨁⨁◯◯ Low
Any anemia at the longest follow up-Cohort studies											
2	observational studies	serious ^e^	not serious ^f^	serious ^c^	not serious	none	20/155 (12.9%)	11/172 (6.4%)	RR 2.26 (1.15 to 4.43)	81 more per 1000 (from 10 more to 219 more)	⨁⨁◯◯ Low
Gastrointestinal blood loss at longest follow up—Randomized Controlled Trials											
1	randomised trials	serious ^g^	not serious	serious ^c^	not serious ^h^	none	9/21 (42.9%)	3/22 (13.6%)	RR 3.14 (0.98 to 10.04)	292 more per 1000 (from 3 fewer to 1000 more)	⨁⨁◯◯ Low
Gastrointestinal blood loss—Cohort study											
1	observational studies	serious ^i^	not serious	serious ^j^	serious ^k^	none	26/60 (43.3%)	6/21 (28.6%)	RR 1.52 (0.73 to 3.16)	149 more per 1000 (from 77 fewer to 617 more)	⨁◯◯◯ Very low
Weight-for-age at longest follow up-Randomized Controlled Trials											
3	randomised trials	serious ^l^	not serious ^m^	serious ^n^	not serious ^o^	none	194	362	-	SMD 0.02 SD lower (0.26 lower to 0.21 higher)	⨁⨁◯◯ Low
Height-for-age at the longest follow up-Randomized Controlled Trials											
2	randomised trials	serious ^p^	not serious ^q^	serious ^n^	not serious ^r^	none	185	344	-	SMD 0.07 SD higher (0.15 lower to 0.3 higher)	⨁⨁◯◯ Low
Serum hemoglobin concentration at the longest follow up—Randomized Controlled Trials											
3	randomised trials	serious ^s^	not serious ^b^	serious ^c^	not serious	none	82	168	-	SMD 0.32 SD lower (0.59 lower to 0.05 lower)	⨁⨁◯◯ Low
Serum hemoglobin level—Cohort studies											
2	observational studies	serious ^t^	not serious	serious ^j^	not serious ^u^	none	148	98	-	SMD 0.37 SD lower (0.78 lower to 0.05 higher)	⨁⨁◯◯ Low
Iron deficiency anemia at the longest follow up-Cohort studies											
2	observational studies	not serious	not serious ^f^	serious ^c^	not serious	strong association	20/155 (12.9%)	11/172 (6.4%)	RR 2.26 (1.15 to 4.43)	81 more per 1000(from 10 more to 219 more)	⨁⨁◯◯ Low
Constipation-Cohort study											
1	observational studies	not serious	not serious	serious ^j^	serious ^v^	strong association	7/69 (10.1%)	3/98 (3.1%)	RR 3.31 (0.89 to 12.37)	71 more per 1000 (from 3 fewer to 348 more)	⨁◯◯◯ Very low
Diarrhea-Cohort study											
1	observational studies	not serious	not serious	serious ^j^	not serious	none	21/69 (30.4%)	16/98 (16.3%)	RR 1.86 (1.05 to 33.10)	140 more per 1000 (from 8 more to 1000 more)	⨁◯◯◯ Very low
Neurodevelopment outcome (PDI scores) at the longest follow-Randomized Controlled Trial											
1	randomised trials	not serious	not serious	serious ^j^	serious ^w^	none	160	268	-	SMD 0.18 SD higher (0.02 lower to 0.37 higher)	⨁⨁◯◯ Low
Neurodevelopment outcome (MDI score) at the longest follow up-Randomized Controlled Trial											
1	randomised trials	not serious	not serious	serious ^j^	serious ^x^	none	160	268	-	SMD 0.16 SD higher (0.03 lower to 0.36 higher)	⨁⨁◯◯ Low

CI, confidence interval; RR, risk ratio; and SMD, standardized mean difference. Explanations: ^a^ One of the two randomized trial studies had “some concerns’ for the risk of bias from the Cochrane risk of bias tool (2). ^b^ No statistical heterogeneity was found in the pooled data. I^2^ = 0%. There was clinical heterogeneity in the type of formula and animal milk use. We did not downgrade the grade level for clinical heterogeneity as there is no consensus on the type of formula or animal milk that should be used when the breastmilk is not available and that multiple options are available for infant formula and animal milk in the community. ^c^ All the included studies were from high-income countries. This might limit the applicability of the results to populations from low and middle-income countries. We, however, think that the direction of effect might remain the same if there were eligible studies from low and middle-income countries and the magnitude of the effect might increase against animal milk. ^d^ Results were statistically significant and the confidence interval is fairly narrow around the summary estimate. ^e^ One cohort study had high risk of bias and the second one had some concerns for risk of bias. ^f^ The I2 statistics was 0%. ^g^ Study had “some concerns” for risk of bias based on Cochrane risk of bias tool (2). ^h^ Even though the confidence interval around the summary estimate included 1, the lower limit of the confidence interval was 0.98. ^i^ The study had “high risk of bias” from the ROBINS tool. ^j^ The only included study for this outcome was conducted in high-income country. ^k^ The confidence interval around the summary estimate included 1 and risk of increased or decreased risk cannot be excluded. ^l^ All three studies were randomized trials. One of the three randomized trial studies had ‘high’ and another has “some concerns” for the risk of bias from Cochrane risk of bias tool-2 (ROB 2). ^m^ The overall unexplained statistical heterogeneity based on 12 statistics was 19 %. The visual inspection of the forest plot showed that three of the included studies had an effect in the same direction and around the mean summary estimate. We did not downgrade the grade level for inconsistency for this outcome. ^n^ All but one of the included studies were from high-income countries. This might limit the applicability of the results to populations from low and middle-income countries. ^o^ The overall magnitude of the effect for the weight for age was small (SMD 0.06). This small statistical effect is not meaningful clinically. Moreover, even though the confidence interval included 0, the total sample size from the pooled studies was 1216. We think there was optimal information size (OIS) from the sample size of the pooled studies that if there was a true effect, that should have been picked up by this much of sample size. We, therefore, did not downgrade for imprecision. ^p^ Two studies were randomized trials. One of the two randomized trial studies had “some concerns” for the risk of bias from the Cochrane risk of bias tool-2(ROB 2). ^q^ Unexplained statistically heterogeneity based on 12 statistics was 17% only. ^r^ The overall magnitude of the effect for the weight for age was small (SMD 0.07) and the confidence interval included 0. This is a very small effect clinically. The total sample size in the analysis was 529 which should have been enough to pick a clinically meaningful effect. We, therefore, did not downgrade the level for imprecision. ^s^ The randomized trial studies had “some concerns” for the risk of bias from the Cochrane risk of bias tool-2. ^t^ One of the observational studies had ‘high’ risk of bias and the other had a ‘moderate’ risk of bias from the ROBINS-1 tool. ^u^ Even though the confidence interval around the summary estimate included a null effect, the upper limit was almost toward the threshold of statistical significance. The data from RCTs showed a similar direction of effect and was statistically significant. ^v^ The 95% CI around the summary estimate included 1. The total sample size was 167 which is not large enough to be confident about the summary estimate. ^w^ The overall magnitude of the effect was small (SMD 0.18) and the confidence interval included 0. ^x^ The overall magnitude of the effect was small (SMD 0.16) and the confidence interval included 0.

**Table 5 nutrients-14-00488-t005:** Use of animal milk vs. formula milk: Results of Secondary Outcomes.

Outcome	No. of Studies	Type of Studies	Total Participants	SMD/RR (95% CI)	I^2^
Iron deficiency anemia	2	Cohort	327	RR = 2.26 (1.15, 4.43)	0%
Blood ferritin at longest follow up	1	Cohort	165	SMD = −0.81 (−1.13, −0.49)	NA
	3	RCT	406	SMD = −0.30 (−0.94,0.34)	85%
Hemoglobin concentration in the stool	2	RCT	228	SMD = 0.22 (−0.16, 0.59)	41%
Hemoglobin concentration in the blood	2	Cohort	246	SMD = −0.37 (−0.78, −0.05)	0%
	3	RCT	250	SMD = −0.32 (−0.59, −0.05)	
Serum iron level	1	Cohort	43	SMD = −0.13 (−0.73, 0.46)	NA
Diarrhea	1	Cohort	167	RR = 1.86 (1.05–33.1)	NA
Constipation	1	Cohort	167	RR = 3.31 (0.89, 12.37)	NA
Neurodevelopmental outcome	1	RCT	428	SMD = 0.18 (−0.02, 0.37)	NA

Footnotes: RR, relative risk; and SMD, Standardized mean difference.

## Data Availability

We will keep all data available for review by the editors and peer reviewers and will provide raw data for the general public on request.

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
