# Peer review of "The Effect of Consumption of Animal Milk Compared to Infant Formula for Non-Breastfed/Mixed-Fed Infants 6–11 Months of Age: A Systematic Review and Meta-Analysis"

_nutrients, 2022, doi:10.3390/nu14030488_

Round 1
Reviewer 1 Report
Thank you for the opportunity to review the manuscript titled “The effect of consumption of animal milk compared to infant formula for non-breastfed/mixed-fed infants 6-11 months of age. A systematic review and meta-analysis”.
The manuscript presents an important aspect of neonatal nutrition, but in my opinion it does not present the discussed issues exhaustively.
Major comments:
- The Introduction section should be expanded and much more describe the differences in the composition of human and animal, including cow milk, in much more detail.
The introduction section poorly presents the composition of human and cow’s milk, which is a widely used substitute of human milk. Infant formula milk is based on cow's milk, however the composition of human and animal milk, including cow's milk, is significantly different, which is related to the different needs of newborns and calves for inorganic and organic ingredients. Moreover, the authors should mention the main problems, which are the net result of substation of human milk by animal milk. In the last paragraph, the authors should argue why it was decided to analyze these and not other effects of animal milk consumption.
- The Discussion section should discuss the results obtained by the authors against the background of previous works. The results should be extensively commented on by the authors and discussed against the background of previous scientific reports.
- The conclusions section should be expanded to include the authors' own thoughts.
- Why do the authors decide to consider results with a p value of less than 0.1 as statistically significant (Usually, statistically significant values below 0.05 are taken)?
Minor comments:
- Please add a list of abbreviations.
- Please standardize the font size in table 1 and 2.
- Line 324-5: Please reorganize this sentence, because in this form it is unclear.
- Line 357: Please add references .
- Figure from 2 to 9: Please add the proper reference number (e.g Fuchs 1993 into Fuchs 1993 [19])
- Line 530: “And because” should not start a sentence. Please reorganized this sentence
Reviewer 2 Report
Exclusive breastfeeding means that the infant receives only breast milk. No other liquids or solids are given – not even water – with the exception of oral rehydration solution, or drops/syrups of vitamins, minerals or medicines. Infants should be exclusively breastfed for the first six months of life to achieve optimal growth, development and health.
The American Academy of Pediatrics recommends that infants be exclusively breastfed for about the first 6 months with continued breastfeeding along with introducing appropriate complementary foods for 1 year or longer. Exclusive breastfeeding practice was very low as compared to recommendations of infant and young child practice (IYCF) which recommends children to exclusively breastfeed for the first 6 months of life. Alternatively, the Dietary Guidelines for Americans (2020) states that infants should not consume cow’s milk before age 12 months as their primary milk drink According to the Lancet Breastfeeding 37% infants in aged 6-24 months not receive breast milk, Although the problem of the benefits of breastfeeding and the significant health impact of this type of feeding over formula feeding is very well known and described here but this form of presenting the issue as a systemic review is very valuable.
This project is different than many others because the intervention of interest was the use of animal milk in infants 6-11 months of age compare to mothers milk or formula.. Because of all above recommendation the limitation their material to infants in the age of 6-11 months is very reasonable. Authors use the systematic reviews which offer a number of benefits. Systematic review aim to identify, evaluate, and summarize the findings of all relevant individual studies over a health-related issue, thereby making the available evidence more accessible to decision makers.
The purpose of this systematic review was to summarize the effect of animal milk consumption, compared to infant formula. The definiction of animal milk was establish as a boiled, pasteurized, or unpasteurized, cow milk or goat, buffalo, camel, sheep milk. The selection of databases use for this systematic review is very appropriate and material makes this manuscript a valuable item.
The authors searched Pub- 101 Med, EMBASE, the Cochrane Central Register for Controlled Trials, Web of Science, CIN- 102 HAL, Scopus, and WHO Global Index Medicus. Unfortunately, the number of publishers meeting the criteria was small. Out of 4340 records, only 7 were selected, 4246 were excluding , among others, because the number of parameters tested ( outcome) in the project assumptions was so large. The results of the systematic review are collected and described in Table 2.
Unfortunately, only two works are from the last 5 years (2013, 2015), the rest from the 1980s. Only 4 studies were randomized controlled trials. Authors included studies in which animal milk was the main milk drink as defined by study authors, or more than 50% of the infant’s milk intake was animal milk. The risk factor was assessed separately for the randomized trials and for cohort study. The evaluation parameters were very extensive and included detailed hematological data, and anthropometric evaluation as well as intestinal disorders
The primary outcomes of interest were: any anemia (dichotomous, as defined by authors); gastrointestinal blood loss (dichotomous, based on stool occult testing); Weight- for-age (continuous, kg or Z scores); Length-for-age (continuous, cm or Z scores); and Weight-for-length Z score. The secondary outcomes were: Iron deficiency anemia, serum iron level, serum ferritin level, Stool hemoglobin concentration Blood hemoglobin concentration Serum tri glycerides, diarrhea (defined as >3 loose stools per day, constipation. Nine studies were included: 4 randomized controlled trials (RCT) and 5 cohort studies. All studies, except one, were conducted in high income countries. Presented flow card is very readable. The conclusions are a bit surprising, but maybe this is due to the bias.
